# Skin Bioimpedance Analysis to Determine Cellular Integrity by Phase Angle in Women with Fibromyalgia: A Cross-Sectional Study

**DOI:** 10.3390/biomedicines11123321

**Published:** 2023-12-15

**Authors:** Davinia Vicente-Campos, Sandra Sánchez-Jorge, Luis Martí, Jorge Buffet, Nuria Mendoza-Laiz, David Rodriguez-Sanz, Ricardo Becerro-de-Bengoa-Vallejo, J. L. Chicarro, César Calvo-Lobo

**Affiliations:** 1Faculty of Health Sciences, Universidad Francisco de Vitoria, Pozuelo de Alarcón, 28223 Madrid, Spain; davinia.vicente@ufv.es (D.V.-C.); luis10marti10@gmail.com (L.M.); j.buffet.prof@ufv.es (J.B.); nuria.mendoza@ufv.es (N.M.-L.); 2Faculty of Nursing, Physical Therapy and Podiatry, Universidad Complutense de Madrid, 28040 Madrid, Spain; davidrodriguezsanz@ucm.es (D.R.-S.); ribebeva@ucm.es (R.B.-d.-B.-V.); jlopezch@ucm.es (J.L.C.); cescalvo@ucm.es (C.C.-L.)

**Keywords:** cellular integrity, fibromyalgia, phase angle

## Abstract

Oxidative stress has been proposed as a significant part of the pathogenesis of fibromyalgia, and the phase angle in bioelectrical impedance analysis has been explored as a potential technique to screen oxidative abnormalities. This study recruited 35 women with fibromyalgia and 35 healthy women, who underwent bioelectrical impedance analysis and maximum isometric handgrip strength tests. Women with fibromyalgia showed lower bilateral handgrip strength (right hand: 16.39 ± 5.87 vs. 27.53 ± 4.09, *p* < 0.001; left hand: 16.31 ± 5.51 vs. 27.61 ± 4.14, *p* < 0.001), as well as higher body fat mass (27.14 ± 10.21 vs. 19.94 ± 7.25, *p* = 0.002), body fat percentage (37.80 ± 8.32 vs. 30.63 ± 7.77, *p* < 0.001), and visceral fat area (136.76 ± 55.31 vs. 91.65 ± 42.04, *p* < 0.01) compared with healthy women. There was no statistically significant difference in muscle mass between groups, but women with fibromyalgia showed lower phase angles in all body regions when compared with healthy control women (right arm: 4.42 ± 0.51 vs. 4.97 ± 0.48, *p* < 0.01; left arm: 4.23 ± 0.48 vs. 4.78 ± 0.50, *p* < 0.001; trunk: 5.62 ± 0.77 vs. 6.78 ± 0.84, *p* < 0.001; right leg: 5.28 ± 0.56 vs. 5.81 ± 0.60, *p* < 0.001; left leg: 5.07 ± 0.51 vs. 5.69 ± 0.58, *p* < 0.001; whole body: 4.81 ± 0.47 vs. 5.39 ± 0.49, *p* < 0.001). Moreover, whole-body phase-angle reduction was only predicted by the presence of fibromyalgia (*R*^2^ = 0.264; β = 0.639; F_(1,68)_ = 24.411; *p* < 0.001). Our study revealed significantly lower phase angle values, lower handgrip strength, and higher fat levels in women with fibromyalgia compared to healthy controls, which are data of clinical relevance when dealing with such patients.

## 1. Introduction

Fibromyalgia is a prevalent, chronic, widespread syndrome involving pain, and is often accompanied by other somatic and psychological symptoms, such as fatigue, poor sleep, cognitive difficulties (memory problems, diminished mental clarity, and concentration difficulties), and psychological distress. Fibromyalgia predominantly affects women, and the average age of onset is around 30–35 years. Moreover, patients with fibromyalgia have lower functional capacity and muscle strength compared to healthy individuals [1,2,3,4].

Although relevant findings have been reported in regard to the pathogenesis of fibromyalgia, the exact mechanisms have yet to be identified. Recently, oxidative stress has been proposed as a significant part [5,6], relating pain to the production of *reactive oxygen species* (*ROS*) [7]. *ROS* are highly reactive molecules that can induce oxidative stress and damage cells and tissues. Evidence suggests that in fibromyalgia mitochondrial dysfunction and increased *ROS* production might contribute to peripheral and central sensitization, interference with pain processing, and development of painful peripheral neuropathies [8], leading to decreased pain threshold, which is a characteristic symptom [9].

Furthermore, a relationship has been established between the imbalance of oxidative products, antioxidant defenses, and the severity of the symptoms. An increase in oxidative stress occurs in patients exhibiting more severe symptomatology [10]. In this regard, Karatas et al. [11] also demonstrated lower levels of antioxidants in patients with fibromyalgia compared to healthy subjects, which further exacerbates this imbalance.

The complexity and diversity of symptoms associated with fibromyalgia complicate its objective evaluation. Thus, the use of a criterion based on biomarkers may be an advantage to objectively evaluate the effectiveness of possible therapeutic treatments. Oxidative stress is normally diagnosed through blood work (including measurement of *ROS* biomarkers, which is a sensitive method). However, these methods are invasive, complex, expensive, and dependent on technicians for collection of blood samples and biochemical analysis, which may limit their use in clinical practice [12].

Bioelectrical impedance analysis (BIA) is one of the methodologies used to calculate body composition parameters, and has been extensively studied for its reliability in obtaining phase angle values [13,14]. BIA is a non-invasive technique that measures human body bioelectrical conductivity value [15]. The phase angle (PhA) obtained through BIA has been explored as a potential way to screen for both inflammatory and oxidative abnormalities [16,17,18]. It is measured through the potential difference of a low-voltage alternating electric current introduced into the body. It is dependent on the resistive behavior and the capacitive effect of the cell membranes and other interfaces [19,20,21,22] and has been proposed as an indicator of cellular health. Higher values indicate higher cellularity, cell membrane integrity, and better cell function [23].

Therefore, monitoring oxidative stress and inflammation through PhA in women with fibromyalgia could be an easy, accessible, and economical way to monitor the development of the disease and its symptoms. To the best of our knowledge, this is the first study to describe PhA values in a sample of women with fibromyalgia and to compare them with those of healthy women. This study pioneers the investigation of PhA in the context of fibromyalgia, shedding light on its potential utility as an indicator of oxidative stress.

## 2. Methods

### 2.1. Participants

A total sample of 70 women aged 31 to 69 years were voluntarily recruited to participate in the study. The sample included 35 women who were diagnosed with fibromyalgia and 35 healthy controls (Figure 1). Written informed consent was obtained from all participants after a detailed description of the investigation procedures was provided. The research was approved by the ethics review board (approval code: 8/2022) of Francisco de Vitoria University (Madrid, Spain). All ethical requirements were respected, including those of the Declaration of Helsinki and human rights for biomedical investigation. The research was registered in www.clinicaltrials.gov (NCT05362396).

The inclusion criteria for the fibromyalgia group were (1) women with a medical diagnosis of fibromyalgia at least one year before the start of the study [24], (2) age between 30 and 75 years, and (3) non-smokers. The inclusion criteria for control group were (1) healthy women, (2) not taking any type of medication on a regular basis, (3) age between 30 and 75 years, and (4) non-smokers. Sample size calculation was performed using G*Power 3.1.9.2, while considering the *t*-test family calculation for the mean difference between two independent groups. The calculation was carried out using an α value of 0.05, a power value of 0.80, a two-tailed hypothesis, an allocation ratio of 1, and a large effect size of *d* of 0.80 [25]. According to these data, a total sample size of 52 participants (26 participants for each group) was necessary to achieve an actual power of 0.807. Considering a possible 30% loss to follow-up, a final sample size of 70 participants was recruited (35 women in each group).

### 2.2. Study Design

This cross-sectional case-control study was carried out with patients who have fibromyalgia and a control group, with 1:1 ratio of cases to controls. The study design was based on the STROBE statements (Strengthening the Reporting of Observational Studies in Epidemiology) [26], which ensure quality and transparency in the presentation of observational studies. The cases comprised patients with a confirmed diagnosis of fibromyalgia, who were recruited from the AFIBROM association (the fibromyalgia patient association of Madrid). The controls were selected from healthy individuals affiliated with Francisco de Vitoria University, and were comparable to the case group in terms of age and sex.

All participants were adequately informed on the study procedures three days prior to assessments and signed the informed consent on the same day. On the measurement day, participants were asked to urinate about 30 min before the tests, to refrain from ingesting food or drink in the previous 4 h, to avoid strenuous physical exercise for at least 24 h, and to refrain from consumption of alcoholic or caffeinated beverages for at least 48 h [27,28]. All measurements were carried out during the morning period. The study’s design and procedures are shown in Figure 2.

### 2.3. Phase Angle (PhA) and Body Composition

PhA and body composition analysis was carried out using octopolar multifrequency electrical (BIA) equipment (InBody 720, Biospace, Los Angeles, CA, USA). This BIA device has shown acceptable reproducibility and accuracy for estimating PhA and body composition at a frequency of 50 kHz [29]. The device provides data on total intracellular water (ICW), extracellular water (ECW), impedance (R), and reactance (Xc), as well as data on different segments of the body (trunk, lower (left and right) and upper (left and right) extremities). It also yields data on skeletal muscle mass (SMM), fat-free mass (FFM), soft lean mass (SLM), body fat mass (BFM), body fat percentage (PBF), visceral fat area (VFA), and body cell mass (BCM). The PhA was calculated as follows [27]:PhA=arctan(Xc/R) × 180°/π

During the measurement, the participants were in their underwear and barefoot in an orthostatic position. They held two electrodes with their hands, and their feet were placed on a platform with two other electrodes, based on the producer’s instructions. Right before the beginning of the assessment, all metallic objects were removed. Then, the contact points where the electrodes were placed on the skin were cleaned with hydrophilic cotton, and the patients remained still and quiet in a supine position during the test. 

### 2.4. Maximum Isometric Handgrip Strength

After the bioimpedance test, maximum isometric handgrip strength was evaluated with a Takei digital grip strength dynamometer (T.K.K. 5401, Takei Scientific Instruments Co., Ltd., Niigata, Japan). The participants were instructed to assume a standing position and grasp the dynamometer while exerting their maximum isometric force with their right and left hand (twice for each side with a 60-s rest period between each time). The elbow and wrist were extended at the side of their body during the test. Subsequently, the individuals were directed to execute a flexion of their fingers and maintain the flexed position by engaging in a maximal isometric contraction lasting for at least 5 s. The maximum score obtained between the two attempts (kg) is reported [30,31].

### 2.5. Statistical Analysis

The software Statistical Package for Social Sciences (SPSS) v. 24.0 (IBM Corp.; Armonk, NY, USA) was used to carry out the statistical analyses, with an α error of 0.05. Statistical significance was considered with a *p*-value less than 0.05 and a 95% confidence interval (CI). For quantitative data, the Kolmogorov–Smirnov test was applied to determine the normality of the distribution. Data values were described as the means ± SDs, lower and upper limits for the 95% CI, and mean difference. *t* and *U* statistics were reported for parametric and non-parametric data, respectively.

Comparisons for parametric data were performed with the Student’s *t* test for independent samples following Levene’s significance for equality of variances. Furthermore, comparisons of non-parametric data were performed with the Mann–Whitney *U* test for independent samples. Regarding differences in outcome measurements, effect size was analyzed with Cohen’s *d* and divided into very small (*d* less than 0.20), small (*d* from 0.20 to 0.49), medium (*d* from 0.50 to 0.79), or large (*d* higher than 0.8) effect sizes [25,32].

Finally, a multivariate linear regression model was carried out to predict the whole-body PhA as the main outcome measurement. This was carried out using the stepwise selection method and the *R*^2^ coefficient to indicate the adjustment quality. Descriptive data (age, weight, height, BMI, right and left handgrip strength) and groups were selected as independent variables. The whole-body PhA was selected as the dependent variable.

## 3. Results

### 3.1. Descriptive Data

The total sample comprised 70 women with a mean age of 52.27 ± 6.63 years, height of 163.32 ± 5.99 cm, weight of 66.54 ± 11.68 kg, BMI of 25.00 ± 4.56 kg/m^2^, right and left handgrip strength of 21.96 ± 7.53 kg and 21.96 ± 7.47 kg, respectively). As shown in Table 1, the groups did not present statistically significant differences regarding age (mean difference (95% CI) = −1.74 (−4.90–1.91) years; *p* = 0.275) and height (mean difference (95% CI) = 1.44 (−2.98–2.80) cm; *p* = 0.953). However, there were statistically significant differences in terms of right (mean difference (95% CI) = −11.14 (−13.55–8.72) kg; *p* < 0.001) and left (mean difference (95% CI) = −11.30 (−13.63–8.97) kg; *p* < 0.001) handgrip strength, which were lower for the fibromyalgia group. This group also had greater weight (mean difference (95% CI) = 6.22 (0.79–11.64) kg; *p* = 0.025) and BMI (mean difference (95% CI) = 2.43 (0.31–4.55) kg/m^2^; *p* = 0.025).

### 3.2. Fat, Water, and Muscle Mass Differences between Groups 

Table 2 shows the differences in fat, water, and muscle mass between groups. There were statistically significant differences with a large effect size (*d* = 0.81–0.91) for greater BFM (mean difference [95% CI] = 7.20 [2.98–11.43]; *p* = 0.002), PBF (mean difference [95% CI] = 7.16 [−3.32–11.00]; *p* < 0.001), and VFA (mean difference [95% CI] = 45.11 [21.67–68.54]; *p* = 0.001) in women with fibromyalgia with respect to healthy women. Nevertheless, the remaining outcomes for water and muscle mass did not show any statistically significant difference (*p* > 0.05) between both women with and without fibromyalgia.

### 3.3. Phase Angle Differences between Groups 

The PhA differences between groups are displayed in Table 3. There were statistically significant differences with a large effect size (*d* = 0.91–1.20). The fibromyalgia group had a lower PhA in all body regions, such as the right arm (RA) at 50-kHz (mean difference [95% CI] = −0.54 [−0.78–0.30]; *p* < 0.001), the left arm (LA) at 50 kHz (mean difference [95% CI] = −0.55 [−0.78–0.31]; *p* < 0.001), the trunk (TR) at 250 kHz (mean difference [95% CI] = −1.16 [−1.55–0.77]; *p* < 0.001), the right leg (RL) at 50 kHz-RL (mean difference [95% CI] = −0.53 [−0.80–0.25]; *p* < 0.001), the left leg (LL) at 50 kHz (mean difference [95% CI] = −0.61 [−0.87–0.35]; *p* < 0.001), and the whole body at 50 kHz (mean difference [95% CI] = −0.57 [−0.80–0.34]; *p* < 0.001).

### 3.4. Multivariate Linear Regression Model for Whole Body PhA Prediction

The linear regression model (*R*^2^ = 0.264) indicated that the whole-body PhA could be predicted by the presence of fibromyalgia (*R*^2^ = 0.264; β = 0.639; F_(1,68)_ = 24.411; *p* < 0.001). The remaining independent variables were excluded (*p* > 0.05) from this prediction model, suggesting that, as a dependent variable, the whole-body PhA was not influenced or predicted by age, height, weight, or right or left handgrip strength according to the pre-established F probability (*P*_in_ = 0.05; *P*_out_ = 0.10).

## 4. Discussion

The main finding of this study is that women with fibromyalgia show lower PhA values compared to healthy control women, adjusted for age. To our knowledge, this is the first study to examine PhA in women with fibromyalgia and to show significantly lower values with this pathology. This investigation was exclusively carried out with women, due to the significantly higher prevalence of fibromyalgia among women compared to men. Research has consistently shown that the majority of individuals seeking medical care for fibromyalgia are women [33,34,35], reporting a ratio of approximately 9:1 between women and men.

In patients with fibromyalgia, an important factor related to pain is oxidative stress through the production of *ROS*. They are formed by oxidation at low levels in the body’s cells and tissues, and their concentration is controlled by the immune system [36]. However, if oxidation levels are high, significant oxidative stress may be caused, leading to peripheral and central sensitization, resulting in a lower pain threshold [9]. 

In addition, *ROS* attack the polyunsaturated fatty acids (PUFAs) in the membrane lipids, resulting in lipid peroxidation and loss of fluidity of the membranes. This also leads to changes in membrane potentials and eventually ruptures, releasing the contents of the cell and organelles [5]. In this sense, oxidative stress is thought to play an important role in the pathogenesis of fibromyalgia [2,6,7,37,38,39].

We excluded smokers from both groups due to the potential implications of oxidative stress associated with smoking. A review by Van der Vaar et al. [40] indicated that acute smoke exposure might lead to tissue damage, as evidenced by elevated products of lipid peroxidation and degradation products of extracellular matrix proteins. This review supported the concept that an imbalance between oxidants and antioxidants may play a pivotal role in smokers. In this sense, Zhou et al. [41] showed that the release of *ROS* from smoking can exceed the body’s antioxidant capacity, leading to oxidative stress and tissue damage.

Khan et al. [42] demonstrated elevated biomarkers of inflammation and oxidative stress in smokers. They compared systemic biomarkers of inflammation, oxidative stress, and tissue injury and repair among cigarette smokers, waterpipe smokers, and dual tobacco smokers. The findings highlighted that biomarkers of systemic inflammation and oxidative stress were notably associated with cigarette smoking. While this study encompassed various tobacco types, it is important to emphasize the link between smoking and oxidative stress in smokers. 

Furthermore, Mons et al. [43] conducted a detailed analysis of data from a smoking cessation trial, revealing that systemic oxidative stress observed in cigarette smokers can arise from both direct exposure to oxidants present in cigarette smoke and indirect induction of inflammatory responses resulting from exposure to constituents in cigarette smoke. This underscores the potential for smoking to induce oxidative stress within the body. Considering this scientific evidence, and in an effort to homogenize the sample while minimizing potential confounding factors, we excluded smokers from the study.

BIA has been explored as a potential technique to screen for both inflammatory and oxidative abnormalities, particularly using the measurement of PhA [17]. PhA is a feasible tool for assessing cellular integrity and could be a potential marker of oxidative and inflammatory processes. PhA potentially represents a cost-effective and efficient tool to monitor individuals at risk for disease development or progression [16]. Thus, considering our results, measuring PhA in patients with fibromyalgia could be a very valuable tool to monitor the development or progression of the disease.

Recently, Akamatsu et al. [44] showed how PhA can be an independent useful indicator of muscle quality. They found that muscle quantity and quality were each independently positively correlated with PhA, suggesting that higher PhA indicated higher muscle quantity and quality. These findings indicate that PhA could be a useful index for easily measuring muscle quality, which has been desired when diagnosing sarcopenia.

In this sense, our results show lower strength values (measured by a handgrip strength test), but not lower muscle mass, which is known as dynapenia [31]. These results are in line with those of a recent study by Kapuczinski et al. [31], who identified a loss of muscle function in fibromyalgia (loss of muscle strength and physical performance), but there was no loss of muscle mass. As in our study, muscle mass was studied through bioimpedance. Koka et al. [45] assessed patients with fibromyalgia in terms of sarcopenia using BIA, anthropometric measures, a handgrip strength test, and a gait speed test over 6 m. They observed that muscle strength was lower in patients with fibromyalgia. The reduction in muscle strength in patients with fibromyalgia compared to healthy controls has also been shown in other studies [46,47,48]. 

Since pain occurs with isokinetic movements, studies have reported possible causes of poor muscle performance in patients with fibromyalgia, including pain, the degree of motivation, sleep disorders, and emotional state [45,46,47]. This could explain our results, in which less muscle mass was not observed in patients with fibromyalgia compared to healthy patients, even though they had less muscle strength. In early 2018, the European Working Group on Sarcopenia in Older People (EWGSOP) updated the definition of sarcopenia with new recommendations. A low muscle strength is now considered to be the primary indicator of probable sarcopenia. When a low muscle strength is detected, a sarcopenia diagnosis must be confirmed by the presence of a concomitant low muscle mass. The severity of the disease is evaluated with the study of physical performance [49]. 

These data are very important to develop strategies that delay the loss of muscle mass in these patients as much as possible to avoid the development of sarcopenia in the future. Easy and simple monitoring of the PhA as an indicator of muscle quality could be useful for monitoring the functionality and quality of life of patients with fibromyalgia. The fact that our results show reduced maximum isometric strength in both hands is consistent with previous studies [50,51,52,53]. Handgrip strength is a rapid and easy way to test muscular fitness that offers valuable insights into overall muscular strength and could have potential applications in clinical settings.

Several reasons have been put forth to explain why patients with fibromyalgia exhibit low levels of handgrip strength, including reduced engagement in daily physical activities or the impact of fatigue and pain, which can adversely affect handgrip strength performance [50]. Our results showed higher BMI, visceral fat, and BFM in women with fibromyalgia compared to controls. These results agree with others in showing how patients with fibromyalgia have a significantly increased fat proportion compared to controls [54,55]. In addition, VF levels were significantly higher in women with fibromyalgia.

Epidemiological data show that patients with fibromyalgia have higher prevalence of obesity (40%) and overweight (30%) in multiple studies compared with healthy patients. Several mechanisms have been proposed to explain the relationship between obesity and fibromyalgia, including impaired physical activity, cognitive and sleep disturbances, psychiatric comorbidity, depression, dysfunction of thyroid gland, dysfunction of the GH/IGF-1 axis, and impairment of the endogenous opioid system. However, it is not currently possible to ascertain whether obesity is a cause or consequence of fibromyalgia [56,57].

What is true is that these data give us important information about this population, as it is directly linked to a higher cardiovascular risk. Due to widespread pain, patients with fibromyalgia have a reduced physical activity and a higher sedentary rate [58], and this may be the reason for these results. In this sense, several studies have shown that higher fat values are associated with greater pain severity, greater number of tender points, reduced physical strength and flexibility, and higher symptom burden in fibromyalgia [45,59,60,61]. Taking these results into account, it would be important to highlight the importance of establishing strategies for weight loss in these patients.

In the multivariate linear regression model, whole-body PhA reduction was only predicted by the presence of fibromyalgia (*R*^2^ = 0.264). Therefore, the sample differences regarding weight, BMI, and bilateral handgrip strength were excluded from this prediction model, indicating these sample features did not influence our study findings according to the pre-established F probability (*P*_in_ = 0.05; *P*_out_ = 0.10). To the best of our knowledge, our study represents a significant contribution, as it is the first to show reduced PhA values in women with fibromyalgia compared to healthy control women. This underscores the potential utility of PhA as an indirect tool to assess oxidative stress, cellular integrity, and inflammatory processes in the context of this condition.

Given the acknowledged association between oxidative stress and pain in patients with fibromyalgia, this easily obtainable marker may serve as a straightforward and valuable tool for monitoring disease progression. Furthermore, our study shows an extensive evaluation of body composition in patients with fibromyalgia, including visceral fat, which is regarded as a highly significant cardiovascular risk factor. Our findings revealed higher levels of visceral fat in women with fibromyalgia, a factor that should be considered when implementing treatment strategies for enhancing quality of life and prognosis of these patients.

The major limitations in our study are its cross-sectional nature, the inclusion of only female patients who have primary fibromyalgia (as fibromyalgia mostly occurs among women), and not categorizing the participants according to their exercise habits. Another limitation of the study could be that women with fibromyalgia exhibit higher levels of BMI compared to healthy women, a fact that is supported by various studies [54,55,56,57]. This could potentially influence the comparison of the PhA between the two groups. Nevertheless, in the multivariate linear regression model, the reduction in whole-body phase angle was only predicted by the presence of fibromyalgia.

For future studies, it would be interesting to increase the sample size and incorporate male participants. In addition, it would be pertinent to consider other potential confounding factors that might impact the outcomes, such as physical activity, nutritional status, or comorbidities, even to categorize by BMI to attempt to address the aforementioned potential limitation. Finally, the study centered on PhA and body composition without directly examining the association between these measurements and fibromyalgia symptoms, such as pain or fatigue. Future research could explore these relationships more comprehensively.

## 5. Conclusions

In conclusion, our study revealed significantly lower PhA values in women with fibromyalgia compared to healthy controls. This finding sheds light on the potential utility of PhA as an indirect tool for assessing oxidative stress, cellular integrity, and inflammatory processes in this condition. Women with fibromyalgia also presented higher levels of visceral fat and lower maximum isometric handgrip strength, highlighting the importance of addressing cardiovascular risk factors in this population.

## Figures and Tables

**Figure 1 biomedicines-11-03321-f001:**
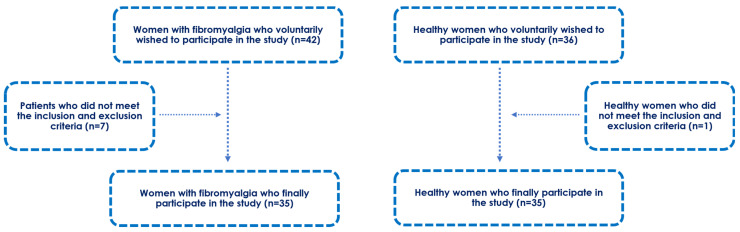
Flow diagram of the participants. Exclusion of seven patients with fibromyalgia due to smoking (3), loss of data (1), and less than one year from the diagnosis (3). Exclusion of one healthy woman due to smoking.

**Figure 2 biomedicines-11-03321-f002:**
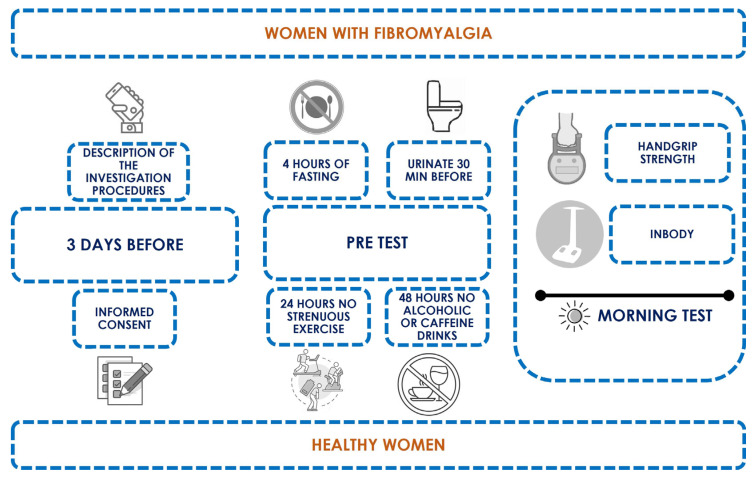
Study design and procedures throughout the study.

**Table 1 biomedicines-11-03321-t001:** Descriptive data for women with fibromyalgia and healthy women.

Descriptive Data	Total Sample (*n* = 70) Mean ± SD (95% CI)	Fibromyalgia (*n* = 35) Mean ± SD (95% CI)	Healthy (*n* = 35) Mean ± SD (95% CI)	Mean Difference (95% CI)	Statistics	*p*-Value
Age (years)	52.27 ± 6.63 (50.68–53.85)	51.40 ± 7.53 (48.81–53.98)	53.14 ± 5.57 (51.22–55.05)	−1.74 (−4.90–1.91)	*t* = −1.100	0.275 *
Height (cm)	163.32 ± 5.99 (161.89–164.75)	163.28 ± 7.36 (160.75–165.81)	163.37 ± 4.33 (161.88–164.86)	1.44 (−2.98–2.80)	*t* = −0.059	0.953 *
Weight (kg)	66.54 ± 11.68 (63.75–69.32)	69.65 ± 13.38 (65.05–74.24)	63.43 ± 8.82 (60.39–66.46)	6.22 (0.79–11.64)	*t* = 2.295	**0.025 ***
BMI (kg/m^2^)	25.00 ± 4.56 (23.91–26.09)	26.21 ± 5.28 (24.40–28.03)	23.78 ± 3.37 (22.62–24.94)	2.43 (0.31–4.55)	*t* = 2.295	**0.025** *
Right HGS (kg)	21.96 ± 7.53 (20.14–23.75)	16.39 ± 5.87 (14.37–18.40)	27.53 ± 4.09 (26.12–28.93)	−11.14 (−13.55–8.72)	*t* = −9.204	**<0.001** *
Left HGS (kg)	21.96 ± 7.47 (20.18–23.74)	16.31 ± 5.51 (14.41–18.20)	27.61 ± 4.14 (26.19–29.04)	−11.30 (−13.63 –8.97)	*t* = −9.690	**<0.001** *

Abbreviations: BMI, body mass index; HGS, handgrip strength; SD, standard deviation. * Student’s *t*-test for independent samples used. For all analyses, *p* < 0.05 (for a 95% confidence interval) was considered as statistically significant (in bold).

**Table 2 biomedicines-11-03321-t002:** Comparison of fat, water, and muscle mass differences between women with and without fibromyalgia.

Outcome Measurements	Fibromyalgia (*n* = 35) Mean ± SD (95% CI)	Healthy (*n* = 35) Mean ± SD (95% CI)	Mean Difference (95% CI)	Statistics	*p*-Value
TBW	31.21 ± 4.03 (29.83–32.60)	31.84 ± 3.06 (30.78–32.79)	−0.62 (−2.22–1.08)	*t* = −0.728	0.469 *
ICW	19.22 ± 2.45 (18.37–20.06)	19.78 ± 1.90 (19.12–20.43)	−0.56 (−1.61–0.48)	*t* = −1.071	0.288 *
ECW	11.99 ± 1.58 (11.45–12.54)	12.05 ± 1.18 (11.65–12.46)	−0.06 (−0.72–0.60)	*t* = −0.179	0.858 *
BFM	27.14 ± 10.21 (23.63–30.65)	19.94 ± 7.25 (17.44–22.43)	7.20 (2.98–11.43)	*U* = 349.000	**0.002** ^†^
SLM	40.03 ± 5.17 (38.25–41.80)	40.90 ± 3.93 (39.55–42.25)	−0.87 (−3.06–1.31)	*t* = −0.796	0.429 *
FFM	42.50 ± 5.48 (40.62–44.38)	43.49 ± 4.20 (42.04–44.93)	−0.98 (−3.31–1.34)	*t* = −0.844	0.401 *
SMM	23.06 ± 3.21 (21.96–24.17)	23.81 ± 2.47 (22.95–24.66)	−0.74 (−2.11–0.62)	*t* = −1.082	0.283 *
PBF	37.80 ± 8.32 (34.93–40.66)	30.63 ± 7.77 (27.96–33.30)	7.16 (−3.32–11.00)	*t* = −3.720	**0.001** *
FFM of Right Arm	2.14 ± 0.39 (2.00–2.28)	2.17 ± 0.31 (2.06–2.28)	−0.02 (−0.20–0.14)	*t* = −0.338	0.736 *
FFM of Left Arm	2.12 ± 0.42 (1.97–2.26)	2.12 ± 0.31 (2.01–2.22)	−0.002 (−0.17–0.17)	*t* = −0.032	0.974 *
FFM of Trunk	19.30 ± 2.58 (18.41–20.19)	19.35 ± 1.85 (18.71–19.99)	−0.04 (−1.12–1.02)	*t* = −0.090	0.928 *
FFM of Right Leg	6.62 ± 1.06 (6.25–6.98)	6.64 ± 0.71 (6.39–6.88)	−0.02 (−0.45–0.41)	*t* = −0.092	0.927 *
FFM of Left Leg	6.61 ± 1.05 (6.25–6.97)	6.61 ± 0.68 (6.37–6.84)	0.0002 (−0.42–0.42)	*t* = 00.001	0.999 *
VFA	136.76 ± 55.31 (117.76–155.76)	91.65 ± 42.04 (77.21–106.09)	45.11 (21.67–68.54)	*U* = 322.000	**0.001** ^†^
BCM	27.52 ± 3.53 (26.31–28.74)	28.34 ± 2.72 (27.40–29.28)	−0.81 (−2.32–0.68)	*t* = −1.082	0.283 *

Abbreviations: BCM, Body Cell Mass; BFM, Body Fat Mass; ECW, Extracellular Water; FFM, Fat Free Mass; ICW, Intracellular Water; PBF, Percent Body Fat; SD, Standard Deviation; SLM, Soft Lean Mass; SMM, Skeletal Muscle Mass; TBW, Total Body Water; VFA, Visceral Fat Area. * Student’s *t*-test for independent samples used. ^†^ Mann–Whitney *U* test applied. For all analyses, *p* < 0.05 (for a 95% confidence interval) was considered as statistically significant (bold).

**Table 3 biomedicines-11-03321-t003:** Comparison of phase angle differences between fibromyalgia and control groups.

Outcome Measurements	Fibromyalgia (*n* = 35) Mean ± SD (95% CI)	Healthy (*n* = 35) Mean ± SD (95% CI)	Mean Difference (95% CI)	Statistics	*p*-Value	Effect Size (Cohen *d*)
50 kHz-RA Phase Angle	4.42 ± 0.51 (4.25–4.60)	4.97 ± 0.48 (4.80–5.13)	−0.54 (−0.78–0.30)	*t* = −4.550	<**0.001** *	*d* = 1.11
50 kHz-LA Phase Angle	4.23 ± 0.48 (4.07–4.40)	4.78 ± 0.50 (4.61–4.96)	−0.55 (−0.78–0.31)	*t* = −4.661	<**0.001** *	*d* = 1.12
250 kHz-TR Phase Angle	5.62 ± 0.77 (5.35–5.89)	6.78 ± 0.84 (6.49–7.07)	−1.16 (−1.55–0.77)	*t* = −5.985	<**0.001** *	*d* = 1.43
50 kHz-RL Phase Angle	5.28 ± 0.56 (5.08–5.47)	5.81 ± 0.60 (5.60–6.01)	−0.53 (−0.80–0.25)	*t* = −3.815	<**0.001** *	*d* = 0.91
50 kHz-LL Phase Angle	5.07 ± 0.51 (4.89–5.25)	5.69 ± 0.58 (5.49–5.89)	−0.61 (−0.87–0.35)	*t* = −4.689	<**0.001** *	*d* = 1.13
50 kHz-Whole Body Phase Angle	4.81 ± 0.47 (4.65–4.98)	5.39 ± 0.49 (5.22–5.56)	−0.57 (−0.80–0.34)	*t* = −4.941	<**0.001** *	*d* = 1.20

Abbreviations: kHz, kilohertz; LA, left arm; LL, left leg; RA, right arm; RL, right leg; TR, trunk. * Student’s *t*-test for independent samples used. For all analyses, *p* < 0.05 (for a 95% confidence interval) was considered as statistically significant (bold).

## Data Availability

The data presented in this study are available on request from the corresponding author. The data are not publicly available due to privacy.

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
