# Peer review of "Skin Bioimpedance Analysis to Determine Cellular Integrity by Phase Angle in Women with Fibromyalgia: A Cross-Sectional Study"

_biomedicines, 2023, doi:10.3390/biomedicines11123321_

Round 1

Reviewer 1 Report

Comments and Suggestions for Authors

General Comments

Since the authors have selected an interesting research topic, their work should be acknowledged. However, the manuscript incurs in conceptual, as well as in structure, writing and conceptual inaccuracies, that significantly decreases its overall merit. It is presented in a simplistic way, and the Introduction and Discussion sections required developing. Moreover, the methods need improvements, like the inclusion of flow chart about the inclusion and exclusion criteria, figures regarding the instruments used and the protocol adopted, better description of the variables studied. On the results, the characteristics of each group are completely different. Apparently, dynamometric differences will appear between different populations. You have not discussed your results in the discussion, are too superficial.  

Reviewer 2 Report

Comments and Suggestions for Authors

This is an interesting article with adequate novelty. However, there are several points that should be improved.

- In the abstract section, the authors should aad an introductory sentences highilighting the literature gap that our study will cover.

- In the abstract section, conclusions of the study should be enriched.

- The introduction is quite short and should be enriched with recent studies on the topic of the study

- At th end of the introduction section, the authors should emphasize the literature gap and the novelty of our study to cover this gap.

- The study design section is very poor and shoulf be enriched with be more details about it.

- In the participants section, concerning inclusion criteria, no smoker women were only included. If there is a reasin for this, the authors should report this issue in the discussion section.

- In the descript data section, a more detailed description of the characteristics of the study population is recommended.

- The above should also be performed for the sections 3.2 and 3.3.

- In the discusion section the authors should include a separate paragraph with the strengths of their studies in order to be more obvious fore the readers.

- In the discusion section the authors should include a separate paragraph with the limitations of their studies in order to be more obvious fore the readers. In this currwent form, the limmitation section is very poor, e.g., a limitation is the small number of the enrolled participants, are there any other techniques beyond phase angle that could be applied in the future.

- The conclusion section is very poor. A paragraph of 5-6 lines is recommended.

- Most of the references are quite old and the authors should be included more reference from the last 5 years.

- Moderate English language editing is required.

Comments on the Quality of English Language

Moderate English language editing is required.

Reviewer 3 Report

Comments and Suggestions for Authors

The authors declare that "The phase angle (PhA) obtained through bioelectrical impedance analysis (BIA) has  been explored as a potential way to screen for both inflammatory and oxidative abnormalities [13–15]. It is measured through the potential difference of a low-voltage alternating electric current introduced into the body. It is dependent on the resistive behavior and the capacitive effect of the cell membranes and other interfaces [16–19] and has been proposed as an indicator of cellular health. Higher values indicate higher cellularity, cell membrane integrity, and better cell function [20]. Therefore, monitoring oxidative stress and inflammation through PhA in women with fibromyalgia could be an easy, accessible, and economical way to monitor the development of the disease and its symptoms. To the best of our knowledge, this is the first study to describe PhA values in a sample of women with fibromyalgia and to compare them with those of healthy women. This study pioneers the investigation of PhA in the context of fibromyalgia, shedding light on its potential utility as an indicator of oxidative stress.

The manuscript is well structured and deals with a topic of potential interest to the scientific community. I just have a few monir suggestions for authors.

As for the introduction, the first part is well written and very clear. The final part, however, could be integrated by adding a more detailed explanation of Bioimpedance, before introducing the phase angle. In this regard the authors could take into consideration the following recent article:

- Sgarro et al., The Role of BIA analysis in Osteopororis risk development: Hierachica clustering approach, Diagnostic (Basel), 2023, 6;13(13):2292.

Regarding the methods section, I would start by describing the sample and, subsequently, provide detailed explanations of the experimental design.

The resolution of the figure seems a little poor.

The "Descriptive data" section, at least in the first part, is a repetition of what is reported in the methods. If possible the authors could provide a table summarizing this section. Otherwise the text could be reshaped to avoid being repetitive.

Reviewer 4 Report

Comments and Suggestions for Authors

Skin Bioimpedance Analysis to Determine Cellular Integrity by Phase Angle in Women with Fibromyalgia: A Case-Control Study

This manuscript covers an important topic .However ,some points need to be addressed

Page

Line

Manuscript

Comments

2

62

The phase angle (PhA) obtained through bioelectrical impedance analysis (BIA)

The research gap has not been fully addressed and previous studies on this topic have not been mentioned

2

78

This case-control study

I think this is a cross sectional study .The authors need to revise this point .

3

95

Figure 1

The figure is to some extent hazy and is not illustrative

What is meant by 7 were excluded ..excluded why ??

4

Figure 2

Th figure does not illustrate wha was done to the healthy individuals

4

Figure 2

There are multiple details in the figure that were not mentioned in the text

5

173

P = .953

All p values were written like that which is not appropriate.

It is better to be written like that  0.953

The authors should illustrated why the study is restricted to women and no males included .

Comments on the Quality of English Language

 Minor editing of English language required

Round 2

Reviewer 2 Report

Comments and Suggestions for Authors

The authors have significantly revised and improved their manuscript.

Comments on the Quality of English Language

Minor editing of English language required

Reviewer 3 Report

Comments and Suggestions for Authors

The authors responded to all my comments.
